# Obesity, Nutrition and Heart Rate Variability

**DOI:** 10.3390/ijms22084215

**Published:** 2021-04-19

**Authors:** Anna Strüven, Christina Holzapfel, Christopher Stremmel, Stefan Brunner

**Affiliations:** 1Department of Medicine I, University Hospital Munich, Ludwig Maximilians University Munich, Marchioninistr. 15, 81377 Munich, Germany; Anna.Strueven@med.uni-muenchen.de (A.S.); christopher.stremmel@med.uni-muenchen.de (C.S.); 2German Center for Cardiovascular Research (DZHK), Partner Site: Munich Heart Alliance, Marchioninistr. 15, 81377 Munich, Germany; 3Institute for Nutritional Medicine, School of Medicine, Technical University of Munich, Ismaninger Strasse 22, 81675 Munich, Germany; christina.holzapfel@tum.de; 4Department of Medicine I, University Hospital Munich, Ziemssenstrasse 1, 80336 Munich, Germany

**Keywords:** nutrition, diet, obesity, weight reduction, heart rate variability

## Abstract

Heart rate variability (HRV) represents the activity and balance of the autonomic nervous system and its capability to react to internal and external stimuli. As a measure of general body homeostasis, HRV is linked to lifestyle factors and it is associated with morbidity and mortality. It is easily accessible by heart rate monitoring and gains interest in the era of smart watches and self-monitoring. In this review, we summarize effects of weight loss, training, and nutrition on HRV with a special focus on obesity. Besides weight reduction, effects of physical activity and dietary intervention can be monitored by parameters of HRV, including its time and frequency domain components. In the future, monitoring of HRV should be included in any weight reduction program as it provides an additional tool to analyze the effect of body weight on general health and homeostasis. HRV parameters could, for example, be monitored easily by implementation of an electrocardiogram (ECG) every two to four weeks during weight reduction period. Indices presumibly showing beneficial changes could be a reduction in heart rate and the number of premature ventricular complexes as well as an increase in standard deviation of normal-to-normal beat intervals (SDNN), just to name some.

## 1. Introduction

Obesity is defined as a pathologically increased body fat mass with a body mass index (BMI) of ≥30 kg/m^2^, whereas overweight describes a pre-stage with a BMI ranging from 25–30 kg/m^2^ [1]. In the year 2000, the World Health Organization (WHO) described obesity as a chronic disease resulting from complex interactions between genetic predisposition, lifestyle, and environmental influences [2]. Worldwide obesity has nearly tripled since 1975. In 2016, 39% of all adults (more than 1.9 billion) were overweight and 13% (over 650 million) were obese. Most of the world’s population live in countries where being overweight and obesity kill more people than being underweight [1].

The development of obesity is complex. Besides diverse influencing factors, a positive energy balance caused by a high energy intake as well as low energy expenditure is the main factor of weight gain [2]. Obesity is associated with several co-morbidities and with a higher mortality rate [3]. It has been identified as a risk factor for several cardiovascular and metabolic diseases, such as diabetes mellitus type 2, hypertension, dyslipidemia, and acute myocardial infarction [4]. It has been shown that physiological variation in the time interval between heartbeats, the so-called heart rate variability (HRV), is pathologically decreased in individuals with cardiovascular diseases [5]. The probability of developing obesity-related cardiovascular diseases as well as the risk of all-cause mortality can be measured by parameters of HRV [6,7,8].

A review of randomised trials in 2018 showed that caloric restriction can improve autonomic regulation and cardiac vagal control as well as lower systolic and diastolic blood pressure [9]. However, weight loss does not directly translate to an increase in HRV and reduced cardiovascular risk. Sekaninova et al. described a dysregulation of autonomic nervous system in anorectic patients leading to higher rate of cardiovascular complications [10,11]. A study of Zhang et al. in 2020 showed that patients after gastric bypass surgery have a disposition to developing orthostatic intolerance. It can be assumed that not only a sympathetic overactivity, but also a general imbalance of autonomic nervous system, including lacking sympathetic activity with thereof resulting loss of vasoconstriction, can arise from weight reduction [12]. Therefore, there is probably more to successful obesity therapy than simply losing weight. A healthy lifestyle including dietary habits itself are also associated with HRV and might be more important than only reduction of body mass.

## 2. Parameters of Heart Rate Variability

### 2.1. Heart Rate Variability and the Autonomic Nervous System

HRV is commonly used for the analysis of activity and balance of the autonomic nervous system (ANS), and can be derived from non-invasive heart rate (HR) monitors [13]. The differences in lengths of beat-to-beat intervals are of special interest as they depict the fluctuation of HR as an answer to several influencing aspects like physical activity or emotional stress, neuroendocrinological processes, and others. These interval variations are also a marker of the capability to regulate internal and external processes [14,15]. A healthy heart is not a metronome. The oscillations of a healthy heart are complex and constantly changing, which allow the cardiovascular system to rapidly adjust to sudden physical and psychological challenges to homeostasis [16].

HR is being controlled by the tenth cranial nerve called vagus nerve in the medulla oblongata, which is a part of our brain stem. Increasing vagal tone leads to a reduction of the HR by inhibiting the sinus node, the heart’s pacemaker [17]. There is a close alliance and communication between brain stem and central autonomic network, with the prefrontal cortex playing a leading role [18]. Direct and indirect pathways involving cingulate and insula cortices, amygdala, hypothalamus, and medulla oblongata link the frontal cortex to autonomic motor circuits responsible for both the excitatory sympathetic and inhibitory parasympathetic effects on the heart [19]. It is conceivable that HRV reflects the overall capacity of the body to deal with on-going requirements for adaptations. Low HRV is associated with a lower capacity for self-regulation of physiological, motional, and cognitive responses and of less effective adaption to environmental stress and demands, including exposure to food stimuli [20,21,22]. Therefore, HRV may act as a biomarker when considering the influence of diet on health-related mechanisms [23].

### 2.2. Heart Rate Variability Domains

There are mainly two different groups of HRV parameters that are commonly used to measure ANS activity: time domain and frequency domain. Time domain measures are calculated by examining the intervals between heartbeats or normal-to-normal (NN) segments measured in milliseconds. Mathematical processing of the N variables leads to a large number of HRV variables including the standard deviation of NN intervals (SDNN) and root mean square of successive differences between NN intervals (RMSSD).

Spectral analyses of fluctuations in HR result in frequency domain HRV variables. These include power in a very low frequency range (VLF; ≤0.04 Hz), low frequency range (LF; 0.04–0.15 Hz), high frequency range (HF; 0.15–0.40 Hz) measured in ms^2^ and percentage of consecutive NN intervals more than 50 ms (pNN50) [13,24]. While power in the VLF and LF range is associated with sympathetic nervous system (SNS) activity, the parasympathetic nervous system (PNS) is represented by the HF range [25]. Mathematical calculations of these frequency domain measures, such as the LF/HF ratio, are helpful measurements of ANS activity, as higher LF/HF ratios are possibly reflecting high SNS activity [26] (Table 1).

## 3. Heart Rate Variability in Individuals with Obesity

Several studies have demonstrated an inverse association of weight gain and obesity with alteration of HRV parameters [27,28,29]. There are various determinants which can influence HRV in individuals with obesity including, e.g., co-morbidities, dietary habits, physical activity, emotional stress, and genetic factors [5,30,31,32].

### 3.1. Glucose Levels and Insulin Resistance

Silva and colleagues showed a statistically significant correlation between high fasting plasma glucose levels and reduced parasympathetic modulation, meaning reduced heart rate recovery after cardiopulmonary exercise testing [33]. Similarly, Kiviniemi et al. showed an association between glucose levels and reduced parasympathetic modulation [34]. In 2020, Oliveira et al. examined 64 middle-aged persons with obesity and found that among the cardiovascular risk variables studied, insulin resistance and waist circumference showed the greatest influence on cardiac autonomic modulation, as they were negatively associated with HF power, a marker of PNS activity [35].

### 3.2. Body Fat

Vagal modulation was also inversely associated with body fat percentage, high body mass, and waist circumference [34,36,37,38]. Individuals who were overweight had sympathovagal imbalance due to increased sympathetic activity associated with visceral fat [39]. RMSSD as well as LF power are negatively correlated with fat percentage and waist-to-hip ratio [23].

### 3.3. Binge Eating Disorders

In 2019, Godfrey et al. investigated the correlation between HRV and emotion regulation in individuals with BMI ≥ 30 kg/m^2^ with binge eating disorder, meaning eating large amounts of food (overeating, OE) and those feeling unable to resist eating (loss of control in eating, LOC). They found that there was a significantly negative association between SDNN and LOC as well as resting LF/HF power and OE. There was a statistical trend for a negative correlation of resting RMSSD with LOC. No significant association was found between HRV during mental stressors (solving math problems) in individuals with LOC or OE. The ANS activity associated with LOC and OE and HRV might be a feasible marker of emotion regulation in people with binge eating disorder [13].

### 3.4. Exercise Training in Individuals with Obesity

Most studies, which investigated training-associated effects on HRV in individuals with obesity, focused on weight and BMI, and did not consider other clinically relevant measures [40,41,42]. Weight reduction via exercise training in persons with obesity has been shown to improve HRV variables by modulation of vagal activity [29,43,44].

In individuals with severe obesity, moderate to vigorous physical activity (MPVA, at least 150 min per week in bouts of at least 10 consecutive minutes) has been shown to be one of the main factors that affect the alteration of autonomic function (beside insulin resistance, central adiposity, and sedentary time) [35].

In the same line, the observational CARDIA study demonstrated that improved cardiorespiratory fitness is a stronger and independent determinant of higher HRV than weight loss [45].

## 4. Effects of Diet on Heart Rate Variability

It is speculated that HRV can be used to indicate the potential health benefits of food items. Several aspects of diet have an effect on HRV.

### 4.1. Energy and Macronutrient Intake

#### 4.1.1. Macronutrient Intake

Food composition has substantial influence on the cardiac autonomic nervous system. In 2003, Tentolouris compared the impact of carbohydrate- and fat-rich meals in 15 obese and lean women. Before food intake, obese women had higher sympathetic activity than thin controls (higher values of low-to-high frequency ratio [LF/HF], 1.52 ± 0.31 versus 0.78 ± 0.13, *p* = 0.04; as well higher plasma norepinephrine levels, 405.6 ± 197.9 versus 240.5 ± 95.8 pg/mL, *p* < 0.0001). After the carbohydrate-rich meal, a greater increase in LF/HF and in plasma norepinephrine levels was observed in lean compared to obese women (1.21 ± 0.6 versus 0.32 ± 0.06, *p* = 0.04; and 102.9 ± 35.4 versus 38.7 ± 12.3 pg/mL, *p* = 0.01, respectively), while no differences were observed after the fat-rich meal [46].

Oliveira et al., in 2021, showed an impaired cardiac autonomic function in subjects with higher fasting blood glucose (defined as >90.5 mg/dL) at fasting as well as after a carbohydrate load (*p* < 0.05). SDNN, LF, and LF/HF increased and HF decreased after consumption of dextrose compared with fasting (*p* < 0.05) [47]. Autonomic modulation following carbohydrate ingestion (CI) consisting of 600 kcal, carbohydrate 78%, protein 13%, and fat 8% and postural stress in standing position (PS) were investigated in a study by Cao et al. (2016) in 14 healthy men and 21 age-matched postmenopausal women (age: 65.0 ± 2.1 vs. 64.1 ± 1.6 years), with intact insulin response. In response to CI and PS, mean arterial pressure maintained stable, and heart rate increased in women and men in the lying and standing positions. Following CI (60, 90, and 120 min postprandially) in PS, systolic blood pressure variability increased by 40% in men (*p* = 0.02) with unchanged HRV parameters; in contrast, in women, HF power halved (*p* = 0.02) with unaltered systolic blood pressure variability. In summary, CI induces sex-specific vascular sympathetic activation in healthy older men and parasympathetic inhibition in healthy older women [48].

Over two days, Lima-Silva et al. examined the impact of a low-carbohydrate intake in those who had been exercising. Compared with a high-carbohydrate intake, the low-carbohydrate diet increased LF and decreased HF power. There were, however, no differences in HR or RR intervals [49].

#### 4.1.2. Caloric Restriction and Energy Expenditure

Sjoberg et al., in 2011, showed an increase in HRV as indicated by low-frequency power (LFP), SDNN, and RMSSD in overweight adults with type 2 diabetes after weight loss (*p* ≤ 0.03) [50].

In 2010, the life-extending effect of caloric restriction (CR) had been investigated in the CALERIE trial [51]. Participants were randomized in four different study arms: (I) control group, (II) CR group with 25% decrease in energy intake, (III) calory restriction and energy expenditure (CREX) group with 12.5% decrease in energy intake plus 12.5% increase in energy expenditure (EX), or (IV) low caloric diet (LCD) group aiming for 15% weight reduction followed by weight maintenance. After six month, HR and SNS index decreased and PNS index increased in all intervention groups but reached significance only in the CREX group. The results therefore suggest that weight loss improve SNS/PNS balance, especially when CR is combined with exercise.

A study in 2003 investigated the influence of diet-induced weight loss on autonomic activity during sleep. Seventeen severely overweight women (BMI > 40 kg/m^2^) underwent a 3-month weight loss program and showed an average weight loss of 10 percent of the starting weight. There was a significant reduction in mean heart rate after weight loss compared with baseline (*p* < 0.001) and also a considerable increase in the parasympathetic parameters of HRV, in particular HF, LF, and VLF power as well as SDNN, RMSSD, and pNN50 [52].

Another study in 2012 found that after caloric restriction for an average of seven years, HR was lower and several measures of HRV values were significantly higher. In fact, HRV after some years of CR diet was comparable to the norms for those 20 years younger. The overall impression is that weight gain adversely influences HRV, although this effect may be reversible with weight loss and/or dietary restriction [53].

### 4.2. Fatty Acids

Certain components of HRV may be influenced by the type of dietary fat, at least in the longer term. In 2013, Sauder and colleagues reported that in healthy adults with elevated triglycerides, supplementation of high amount of omega-3 fatty acids (3.4 g/d eicosapentaenoic acid and docosahexaenoic acid) resulted in a 9.9% increase of RMSSD and 20.6% increase of total power, showing an improvement of autonomic function [54].

The health-promoting effects of omega-3 fatty acid intake have been frequently studied. For example, Baumann et al. gave 20 obese children (mean body mass index percentile: 99.1; mean age: 11.0 years) a daily supplementation of 400 mg eicosapentaenoic acid (EPA) and 120 mg docosahexaenoic acid (DHA) for at least three months and compared their HRV response with the one of 94 normal-weight children. Time domain HRV parameters as indicators of vagal function were significantly lower in obese subjects than in the control group, but HR was higher (SDNN = −34.02%; RMSSD = −40.66%; pNN50 = −60.24%; HR: = +13.37%). After omega-3 fatty acid supplementation, time domain HRV parameters and HR of obese patients were similar to the values of healthy controls (SDNN interbeat intervals: −21.73%; RMSSD: −19.56%; pNN50: −25.59%; HR: +3.94%) [55].

There is a general impression that omega-3 intake results in greater parasympathetic activity. Christensen and Schmidt concluded that in five different studies, dietary n-3 polyunsaturated fatty acids (PUFA) levels and n-3 PUFA supplementation are related to improved HRV. The findings suggested that an increased parasympathetic regulation of heart activity occurred [56].

Billman reviewed the evidence that the intake of omega-3 fatty acids influenced cardiac rhythm. He concluded that supplementation with n3-PUFAs affects ion channels and calcium-regulatory proteins. Immediately after metabolic processing, there is a direct effect of the fatty acids on ion channels, while after a longer period, when the incorporation of the fatty acids into the cell membrane has proceeded, cardiac electrical activity changes. HR is reduced and HRV increases, reflecting alterations in the intrinsic pacemaker rather than regulation by the activity of the ANS [57].

### 4.3. Micronutrients

Vitamins are found to be another influencing factor of HRV. Celik et al. examined sixty adolescents with Vitamin B12 deficiency (Vitamin B12 ≤ 200 pg/mL, mean age 14.4 ± 1.72 years) and 40 healthy controls (mean age 13.4 ± 1.86 years). LF, HF (*p* < 0.001), and RMSSD (*p* = 0.04) were significantly lower in the B12 deficient patients [58]. The impact of a supplementation of 400 mg magnesium per day on sympathovagal function was investigated in a randomized, placebo-controlled study with 100 participants over a time span of 90 days in 2016. In the treatment group, pNN50, as a marker of vagus activity, significantly increased and LF/HF ratio decreased while no positive changes appeared in the control group [59]. 

A review by Lopresti in 2020 investigated cross-sectional and interventional studies about the relation between HRV and vitamin B-12, C, D, and E and the minerals magnesium, iron, zinc, and coenzyme Q10 as well as a combination in the form of a multivitamin-mineral formula. Although this study does not allow a clear inference about the effects of these micronutrients on HRV due to the sparse number, the heterogeneity, and the relatively low power of available scientific studies on these correlations, it can be strongly assumed that deficiencies in vitamin D and B-12 are associated with impaired HRV, and zinc intake, especially during pregnancy, can have positive implications on HRV in infants until the age of five years [60].

### 4.4. Sodium

Sodium intake is also linked to HRV. Buchholz et al., in 2003, showed by means of 17 salt-sensitive young men and 56 salt-resistent controls that stress-induced heart rate increases more in the salt-sensitive than salt-resistant group (*p =* 0.01). Additionally, the salt-sensitive men, in comparison to control, showed significantly reduced time domain-based heart rate variability RMSSD at baseline (*p =* 0.01) and during an 8 min mental stress test (*p =* 0.05). Salt-sensitivity was defined as a > 3 mmHg reduction of systolic blood pressure after 14 days of low-salt diet containing daily intake of 20 mmol sodium, 20 mmol chloride, 60 mmol potassium, and 20 mmol calcium in addition to a daily supplement of 22 tablets of Slow Sodium (10 mmol sodium chloride per tablet). Diastolic blood pressure response was also higher in the salt-sensitive group (*p =* 0.05) [61].

### 4.5. General Antiarrhythmic Effects of Nutrition

There is a growing body of literature about the central role of dietary components in the prevention of cardiac arrhythmias, especially fatal arrhythmias (ventricular arrhythmias), but little evidence is available regarding the protective effects of dietary patterns and lifestyle on premature ventricular complexes (PVCs) and HRV [62,63].

Results from randomized controlled trials suggest that the sufficient supplementation of omega-3 PUFA may reduce the overall number of PVCs, also decreasing their severity [64,65]. Moreover, omega-3 PUFAs have also been hypothesized to be linked to HRV, supporting their protective role in subjects at high risk for arrhythmic events and sudden cardiac death (SCD) [66]. In contrast with substances that may play a role in preventing PVCs, some food contents may act as triggers. Some studies have suggested that alcohol and caffeine consumption may be linked to cardiac performance. Evidence indicates the existence of an association between different doses of alcohol consumption with HRV and cardiac ectopic beats, while the role of caffeine is still disputed [67,68,69]. In a randomized, single-blind study in 2010, the effect of red wine, ethanol, or water on HRV was observed in healthy subjects aged 24–57 years (each 50% women or men). One alcoholic drink increased blood alcohol concentration to 36 ± 2 mg/dL (mean ± standard deviation), and two drinks to 72 ± 4 mg/dL (red wine) and 80 ± 2 mg/dL (ethanol). Red wine quadrupled plasma concentration of antioxidant polyphenol resveratrol (*p* < 0.001). When compared with respective baselines, one alcoholic drink had no effect on HR or HRV, whereas two glasses of red wine or ethanol increased HR (red wine, +5.4/min ± 1.2/min; and ethanol, +5.7/min ± 1.2/min; *p* < 0.001) decreased total HRV by 28–33% (*p* < 0.05) and high-frequency spectral power by 32–42% (parasympathetic HR response), and increased LF power by 28–34% and the ratio of LF to HF by 98–119% (sympathetic HR response) (all, *p* ≤ 0.01). All in all, alcohol shows dose-dependent effects on HRV, without significant difference between red wine and ethanol. Drinking plain water resulted in a reduction of HR [70]. In 1993, Aparichi et al. showed that tobacco consumption leads to a higher heart rate and higher number of PVCs as well as supraventricular arrhythmias, an effect which can be partly reversible after nicotine abstinence [71]. A questionable aspect in non-organic food is the arrhythmic effect of plant protection products and nitrates from preserved food. A study in 2015 showed an increase in first sympathetic tone, then prolongation, in a parasympathetic period leading to a prolongation of spread of excitaion in the heart (QT), possibly followed by torsade de pointes or even ventricular fibrillation and sudden cardiac death after intoxication by organophosphorus insecticide chlorpyrifos, a widely used plant protection substance known to be injurious to health [72].

A current study by Reginato et al. investigated differences in the presence or absence of PVC and supraventricular premature complexes (SPVCs) in several dietary patterns and found that, in the multivariable analysis, there was a statistically significantly lower number of PVCs and SPVCs in those consuming a higher amount of fruits (*p* = 0.044) and grain-based products (*p* = 0.001) compared with others. The intake of sugary foods worsened arrhythmias (*p*-value 0.013). Regardless of the source (animal- or plant-based), protein food consumption was significantly and inversely proportional to SPVCs (*p*-value < 0.001). Similar results have been obtained from the analysis of the general frequency of food consumption, while they did not differendiate between the sources of protein meaning coming from fish, plant, or milk products. This study also showed a significant direct association between waist circumference and PVCs and an inverse correlation between fruit intake and PVCs. Higher consumption of sweets, sugars, and refined grain products led to an increase of the number of PVCs [63]. From previous investigations, limited evidence is available regarding the association between protein food and cardiac electrical activity, since earlier studies have concentrated mainly on fish intake and cardiac rhythm, not on another type of protein food. Results of such studies suggested a beneficial role of omega-3 PUFA in fish against cardiac arrhythmias, primarily pointing out the kind of fatty acids being consumed [73,74] (Table 2).

### 4.6. General Nutrition Recommendation

Finally, we would like to name the current official recommendations of the German food society (Deutsche Gesellschaft für Ernährung—DGE) and German adiposity society (Deutsche Adipositas-Gesellschaft—DAG) for a healthy and complete nutrition and life style: food should be varied and mainly plant-based, five portions of fruits and vegetables should be consumed every day, whole-grain products should be preferred to simple carbohydrates, protein intake should include a daily portion of milk products and a fish meal once to twice a week. Consumption of meat is not generally recommended, but if included in the dietary plan, it should not go beyond 300 to 600 g (g) per week. Polyunsaturated fatty acids should be used instead of animal fats, intake of sugar should be as low as possible, and the use of salt as a condiment as far as possible should be replaced by herbs. The fluid intake should include around 1.5 L of plain water and preferably only a little amount of sweetened or alcoholic drinks, meals should be taken in peace, and last but not least, regular exercise is considered health-promoting. The American Heart Association (AHA) emphasizes the importance of a normal BMI ranging from 18.5 to 25 kg/m^2^ [75,76,77,78,79] (Table 3).

## 5. Conclusions

In summary, although there has been a limited number of systematic reviews, there is a series of reports that relate various indices of HRV to different lifestyle factors. Additionally, reduced physical activity and an imbalanced diet are clearly linked to a reduced PNS modulation. As HRV is a measure of the body’s capability to react to internal and external challenges in the sense of homeostasis, it is not surprising that any decrease in HRV might lead to higher morbidity and mortality. Therefore, HRV is an easily accessible marker of a healthy lifestyle. Especially, programs that tackle obesity should include this parameter, as it reacts to many physiological parameters beyond weight loss and provides good feedback for comprehensive therapy success in a short- and long-term range.

## Figures and Tables

**Table 1 ijms-22-04215-t001:** Parameters of Heart rate variability.

Method	Variability Measure	Measurement Unit	Definition and Explanation	Indicator of …	Assignment as Part of the Autonomic Nervous System	Recommendation for Reporting Time
Statistical	SDNN	ms	Standard deviation of NN intervals	Total variability	No clear assignment	
Statistical	RMSSD	ms	Root mean square of successive differences of all consecutive NN intervals	Short-term variability	PSY	
Statistical	pNN50	ms	Percentage of mean number of successive normal sinus (NN) consecutive RR-Intervals exceeding 50 ms	Total variability	PSY	
FFT and autoregressive model	VLF	ms^2^	Very low frequency power: power spectral density in the frequency range from 0.003 to 0.04 Hz	No clear assignment	SY	
FFT and autoregressive model	LF	ms^2^	Low frequency power: power spectral density in the frequency range from 0.04 to 0.15 Hz	No clear assignment	SY > PSY	≥5 min
FFT and autoregressive model	HF	ms^2^	High frequency power: power spectral density in the frequency range from 0.15 to 0.40 Hz	No clear assignment	PSY	≥5 min
FFT and autoregressive model	LF/HF	k. E.	Ratio of sympatho-vagal balance; measurement of interaction between SNS and PNS	No clear assignment	SY and PSY	≥5 min

HRV—Heart rate variability, FFT—Fast Fourier Transformation, SNS—sympathetic nervous system, PNS—parasympathetic nervous system, SY—Sympathicus, PSY—Parasympathicus, NN—normal-to-normal beat interval, ms—milliseconds, min—minutes, Hz—hertz unit, k.E.—keine Einheit.

**Table 2 ijms-22-04215-t002:** Obesity and heart rate variability, assembling of studies.

References	Sample	Design	Outcome
Godfrey 2019	28 persons with obesity	HRV and emotion regulation in binge eating disordered with BMI ≥ 30 kg/m^2^	Statistically significant association between SDNN, resting low and high frequency domain and overeating, association between resting RMSSD and binge eating -> HRV as a feasible marker of emotion regulation
Chen 2019	2316 middle-aged persons with obesity	BMI, HRV und graded exercise test duration at randomization and after 20 years	Higher waist circumference and higher measures of adiposity as well as lower level of cardiorespiratory fitness leading to lower RMSSD, higher waist circumference meaning lower SDNN
Oliveira 2020	64 middle-aged persons with obesity	Risk Factors influencing cardiac autonomic function in persons with obesity	Insulin resistance and waist circumference showed the greatest influence on cardiac autonomic modulation of obese as negatively associated with high frequency power, representing the parasympathetic activity
Jonge 2010	48 persons with over-weight	Life extending effect of caloric restriction (CR)	Randomization in 3 groups: control group, CR group with 25% decrease in energy intake, calory restriction and energy expeniture (CREX) group with 12.5% CR plus 12.5% increase in energy expenditure (EX), or low calory diet (LCD) group aiming 15% weight reduction. After six months heart rate (HR) and sympathetic nervous systeme (SNS) index decreased and parasympathetic nervous systeme (PNS) index increased in all intervention groups but reached significance only in CREX. Heart rate and SNS index increased and PNS index decreased after having meal in all intervention groups. Conclusion: weight loss improved SNS/PNS balance especially when CR is combined with exercise
Chintala 2015	40 controls, 40 persons with over-weight	Correlation visceral fat and HRV	Overweight individuals had sympathovagal imbalance due to increased sympathetic activity associated with visceral fat
Poirier 2003	17 severe overweight women (BMI > 40 kg/m^2^)	Effect of diet induced on HRV in severe obese women	An average weight loss of 10 percent showed a significant reduction in mean heart rate (HR) and notable increase in several parasympathic parameters (HF, LF, VLF power, SDNN, RMSSD and pNN50) within three months
Stein 2012	22 adult caloric restriction individuals, 20 controls eating Western diets	Association between HRV and caloric restriction (CR)	After 7 years of CR, lower HR and higher HRV—comparable to the norm for those 20 years younger—occured. Conclusion: weight gain adversely influences HRV, although this effect may be reversible with weight loss and/or dietary restriction.

HRV—heart rate variability; BMI—body mass index; SDNN—standard deviation of NN intervals; RMSSD—root mean square of successive differences between NN intervals; CR—caloric restriction; CREX—calory restriction and energy expeniture; EX—energy expenditure; LCD—low calory diet; HR—heart rate; SNS—sympathetic nervous systeme; PNS—parasympathetic nervous systeme; kg—kilogram; m^2^—square meters; pNN50—percentage of mean number of successive normal sinus (NN) consecutive RR-Intervals exceeding 50 ms.

**Table 3 ijms-22-04215-t003:** General nutrition recommendation of DGE, DAG and AHA for a healthy and complete nutrition and life style.

No.	General Nutrition Recommendation of DGE, DAG and AHA
1.	Varied, mainly plant based food
2.	5 portions of fruit/vegetables per day
3.	Predominantly whole grain products, fewer simple carbohydrates
4.	Daily intake of milk products, fish meal once to twice per week, meat as little as possible, maximum 300–600 g per week
5.	Preferably polyunsaturated fatty acids instead of animal fats
6.	Salt mainly replaced by herbs, Sugar as little as possible
7.	1.5 L of plain water per day
8.	Gently cooking
9.	Meals taken in in peace
10.	Regular physical exercise

## Data Availability

Not applicable.

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
