# Peer review of "Obesity, Nutrition and Heart Rate Variability"

_ijms, 2021, doi:10.3390/ijms22084215_

Round 1
Reviewer 1 Report
IJMS-1176731 review
This review sets out to discuss measurement of heart rate variability (HRV) as a tool to follow changes in autonomic nervous system activity. It aims to summarize the effect of weight, exercise and nutrition on HRV and introduce the concept of HRV as a tool to analyze overall health. In general the review is comprehensive in its treatment of the topic.
I feel that a few changes would improve the manuscript:
The introduction mentions “monitoring of HRV should be included in any weight reduction program”. Perhaps the authors can include a section with recommendations on how to monitor HRV during weight reduction and which indices reflect beneficial changes.
A suggestion is to consider including a short discussion on orthostatic intolerance in patients after gastric bypass surgery (Obes. Sci Pract. 2019 6(1) 76-83 PMID 32128245-is an example of a case series describing this) since this seems to be a relatively common finding and may be relevant to the discussion in this review.
Section 4.5 General antiarrhythmic effects of nutrition was interesting, particularly discussion of the negative effects of commonly ingested substances on PVC. The authors should consider expanding this a little more to include discussion of the effects other modern ingested substances (ie nitrates from preserved food, high fructose corn syrup, nicotine etc).
Section 4.6 General nutrition recommendation may be better as a figure, table or supplement and not as part of the main body of the text.
Minor comments:
Line 57 It has been shown, that physiological…Comma not necessary and interrupts flow of sentence
Line 86 and medulla oblongata are linking…Should read “and medulla oblongata link”
Line 144 The ANS activity was associated with LOC…Should read “The ANS activity associated with LOC…”
Line 154 one of the main factors, that affect…Comma not necessary and interrupts flow of sentence
Line 155 sedation time…This does not sound correct. Sedentary time might be better
Line 167 in each 15…Can omit “each”
Line 178 carbohydrate overload…Maybe replace with “after a carbohydrate load”
Line 179 excessive consumption of dextrose…Maybe omit excessive
Line 204 aiming 15% weight reduction…Maybe “aiming for 15% weight reduction”
Line 224 in the longer term: …Replace with “in the longer term.” (Replace colon with period)
Line 225 supplementation with a high amound of…amound -> amount
Line 226 and 227 9,9% and 20,6%...Use comma or decimal point consistently throughout
Line 228 The health-promoting effect… effect -> effects
Line 246 Immediately after metabolization… Maybe change metabolization -> metabolic processing
Line 263 and 264 the relation between HRV and vitamin B12, C, D and E …and coenzyme Q…Was the study supplementing these vitamins and minerals or measuring levels in blood or intake?
Line 270 sprout…Maybe change sprout -> infants
Line 282 response war also higher … war -> was
Line 293 Together with substances that may play a role…Maybe instead put “In contrast with substances that may play a role in preventing PVCs”
Line 315 The intake of sugary foods harmed… Maybe replace “harmed” with “worsened”
Line 348-349 People with obesity often suffer from impaired glucose tolerance…Consider removing this sentence as it is a little bit obvious and does not help with the conclusion.
Author Response
Dear reviewer,
thank you very much for your precise suggestions for improvement for our review on Obesity, nutrition and HRV.
As a recommendation for monitoring HRV parameters and potential indices presumibly improving while and after weight reduction we added the following text passage (after line 35-37 of original review version):
”HRV parameters could for example be monitored easily by implementation of a thirty minute ECG every two to four weeks during weight reduction period. Indices presumibly showing beneficial changes could be a reduction in heart rate and the number of premature ventricular complexes as well as an increase in standard deviation of normal-to-normal beat intervals (SDNN), just to name some.”
We implemented your proposal for a section on orthostatic imbalance after bariatric surgery ref erring to the recommended study of Zhang et al. 2020 (new citation 12). It now reads as follows: “A study of Zhang et al. in 2020 showed that patients after gastric bypass surgery have a disposition to developing orthostatic intolerance. It can be assumed that not only a sympathetic overactivity but also a general imbalance of autonomic nervous system including lacking sympathetic activity with thereof resulting loss of vasoconstriction can arise from weight reduction.” (Starting in line 66 of original review).
Section 4.5. we added a passage on arrhythmic effects of smoking and plant protection substance.
“In 1993 Aparichi et al. showed that tobacco consumption leads to a higher heart rate and higher number of PVCs as well as supraventricular arrhythmias, an effect which can be partly reversible after nicotine abstinence [71]. A questionable aspect in non-organic food is the arrhythmic effect of plant protection products and nitrates from preserved food. A study in 2015 showed an increase in first sympathetic tone, then prolongation in parasympathetic period leading to a QT prolongation possibly followed by torsade de points or even ventricular fibrillation and sudden cardiac death after intoxication by organophosphorus insecticide chlorpyrifos, a widely used plant protection substance known to be injurious to health [72].”
We created a table for general nutrition recommendation (section 4.6., starting in line 310 of original review).
Table 3
General nutrition recommendation of DGE, DAG and AHA for a healthy and complete nutrition and life style
|
1. |
Varied, mainly plant based food |
|
2. |
5 portions of fruit/vegetables per day |
|
3. |
Predominantly whole grain products, fewer simple carbohydrates |
|
4. |
Daily intake of milk products, fish meal once to twice per week, meat as little as possible, maximum 300-600 g per week |
|
5. |
Preferably polyunsaturated fatty acids instead of animal fats |
|
6. |
Salt mainly replaced by herbs, Sugar as little as possible |
|
7. |
1,5 liters of plain water per day |
|
8. |
Gently cooking |
|
9. |
Meals taken in in peace |
|
10. |
Regular physical exercise |
The review of Lopresti 2020 included several studies viewing either the amound of intake or the deficiencies (blood levels).
The minor comments we changed as recommendet. The new text passages are highlighted in blue colour.
With best regards,
Anna Katharina Strüven

Reviewer 2 Report
This is an excellent review on obesity, nutrition and heart rate variability that covers all relevant topics.
General comments:
Line 119 ff: This paragraph explains the correlation between increased glucose levels, insulin resistance, body fat, obesity and reduced parasympathetic modulation as measured by HRV.
Here, the decisive molecules come in and therefore your review fits within the scope of the journal.
Line 222 ff: Very instructive information on fatty acids: the effects of omega-3-fatty acids are impressive and seem to be highly relevant.
Line 252 ff: This important chapter on the impact of micronutrients is concise and conclusive. The information on salt-sensitive and salt-resistant persons is definitely a must read.
Specific suggestions for minor changes and open questions:
Line 58: "The probability of developing obesity-related cardiovascular diseases as well as the risk of all-cause mortality can be measured by parameters of HRV [6]."
This statement is not covered by the reference cited. Please rephrase or cite additional references that do so.
Line 155: sedation time: do you mean sleep duration? Please explain.
Author Response
Dear reviewer,
thank you very much for your valuable suggestions for improvement for our review on Obesity, nutrition and HRV.
For line 58 (original review version): "The probability of developing obesity-related cardiovascular diseases as well as the risk of all-cause mortality can be measured by parameters of HRV [6]" we additionally cited the review titled “Sympathetic nervous system as a target for aging and obesity-related cardiovascular diseases“ of Balasubramanian 2019 and the study “Heart rate variability measured early in patients with evolving acute coronary syndrome and 1-year outcomes of rehospitalization and mortality“ of Harris 2014
(new citations no 7 and 8).
In line 155 (original review version) we changed “sedation time“ into “sedentary time“, as what is meant is the time of seated activity. Both your recommendations are highlighted in red colour.
With best regards,
Anna Katharina Strüven
